# Multimodal Data Integration to Predict Severe Acute Oral Mucositis of Nasopharyngeal Carcinoma Patients Following Radiation Therapy

**DOI:** 10.3390/cancers15072032

**Published:** 2023-03-29

**Authors:** Yanjing Dong, Jiang Zhang, Saikt Lam, Xinyu Zhang, Anran Liu, Xinzhi Teng, Xinyang Han, Jin Cao, Hongxiang Li, Francis Karho Lee, Celia Waiyi Yip, Kwokhung Au, Yuanpeng Zhang, Jing Cai

**Affiliations:** 1Department of Health Technology and Informatics, The Hong Kong Polytechnic University, Hong Kong SAR, China; 2Research Institute for Smart Ageing, The Hong Kong Polytechnic University, Hong Kong SAR, China; 3Department of Biomedical Engineering, Faculty of Engineering, The Hong Kong Polytechnic University, Hong Kong SAR, China; 4Department of Radiology, Fujian Medical University Union Hospital, Fujian Medical University, Fuzhou 350000, China; 5Department of Clinical Oncology, Queen Elizabeth Hospital, Hong Kong SAR, China; 6Department of Medical Informatics, Nantong University, Nantong 226000, China; 7The Hong Kong Polytechnic University Shenzhen Research Institute, Shenzhen 518000, China

**Keywords:** multimodal data integration, radiomics, dosiomics, nasopharyngeal carcinoma, acute mucositis

## Abstract

**Simple Summary:**

The acute oral mucositis (AOM) is a prevalent side effect of radiation therapy for nasopharyngeal carcinoma (NPC). Severe AOM could impair the survival and quality of life for NPC patients. Accurate method to predict the incidence of severe AOM can aid clinicians in adjusting the treatment plan to improve the outcomes for NPC patients. We integrated multi-modalities, multi-omics and multi-regions data with two methods, integrate the original data or combine data after feature selection. The performance of models using each data integration method with different modalities, types of data and VOIs were analyzed. We developed a best-performing model with mean AUC at 0.81 ± 0.10 to predict the incidence of severe AOM for NPC patients following radiation therapy.

**Abstract:**

(1) Background: Acute oral mucositis is the most common side effect for nasopharyngeal carcinoma patients receiving radiotherapy. Improper or delayed intervention to severe AOM could degrade the quality of life or survival for NPC patients. An effective prediction method for severe AOM is needed for the individualized management of NPC patients in the era of personalized medicine. (2) Methods: A total of 242 biopsy-proven NPC patients were retrospectively recruited in this study. Radiomics features were extracted from contrast-enhanced CT (CECT), contrast-enhanced T1-weighted (cT1WI), and T2-weighted (T2WI) images in the primary tumor and tumor-related area. Dosiomics features were extracted from 2D or 3D dose-volume histograms (DVH). Multiple models were established with single and integrated data. The dataset was randomized into training and test sets at a ratio of 7:3 with 10-fold cross-validation. (3) Results: The best-performing model using Gaussian Naive Bayes (GNB) (mean validation AUC = 0.81 ± 0.10) was established with integrated radiomics and dosiomics data. The GNB radiomics and dosiomics models yielded mean validation AUC of 0.6 ± 0.20 and 0.69 ± 0.14, respectively. (4) Conclusions: Integrating radiomics and dosiomics data from the primary tumor area could generate the best-performing model for severe AOM prediction.

## 1. Introduction

Nasopharyngeal carcinoma (NPC) is a kind of malignant epithelial head and neck (H&N) cancer. It originates from the nasopharyngeal mucosal lining with high incidence in Southeast Asia [1]. Over 95% of NPC cases are non-keratinizing squamous cell carcinoma that is highly sensitive to radiation [2]. Radiotherapy with/without chemotherapy (CRT or RT, respectively) is the primary treatment for NPC patients to achieve disease control. Although advanced radiation strategies such as intensity-modulated RT (IMRT) are widely applied to reduce the recurrence rate of tumor with decreased radiation-induced side effects [3,4], radiation toxicity is still a trouble for NPC patients undergoing radiation therapy [5,6]. The radiation damages DNA and cellular components, which could cause mitotic cell death, apoptosis, and cytokine cascade in the human body. These cascades of events could eventually and inevitably lead to toxicity effects [7]. Most NPC patients suffer from dermatitis, mucositis, dysphagia, and xerostomia during and after RT/CRT. Among these acute toxicity effects, acute oral mucositis (AOM) is the most common painful symptomatic complication for NPC patients [8].

More than 60% of H&N patients have experienced AOM following RT-based anti-cancer therapies. Approximately 65% of these patients have developed severe AOM (≥grade 3) [8]. The AOM is typically characterized by atrophy, swelling, erythema, and ulceration. It often impairs patients’ functional status and quality of life (QoL) [9]. The soreness of AOM initially leads to open-mouth difficulty, which further causes decreased food and water intake, loss of weight, and systematic infection. Patients who have developed severe AOM with painful experiences would receive a reduced dose of chemotherapy. Some patients even tend to break the RT regime. Severe AOM can exacerbate the morbidity of patients, which may finally contribute to worsen QoL and increase mortality [10]. It therefore is necessary to analyze the critical contributors to the oral mucositis.

Previous studies have focused on analyzing one type of data, such as genetic data and clinical variables, for predicting severe AOM. Various factors have been identified to correlate with the incidence and severity of oral mucositis, for instance, genetic background [11,12], dose of RT [13,14], chemo-drugs [15,16,17], and nutritional status [18]. Among these factors, the dose of RT is a critical factor influencing the severity of AOM. Additional chemotherapy, especially with some AOM-associated agents, for instance, alkylating agents and antimetabolites, could exacerbate the events. Besides, AOM patients are more likely to have weight loss during the treatment scheme [19,20]. Saito et al. [20,21] reported that low BMI is a risk factor for severe AOM. Andy et al. [21] indicated that patients with advanced tumors are prone to experience AOM. Moreover, a recent two-stage genome-wide association study [12] showed that four single nucleotide polymorphisms (SNPs) might be correlated with acute mucositis. However, they failed to validate their results in the validation stage. Clinicogenomic variables alone are inadequate to accurately predict the incidence, correlations, and severity of AOM for NPC patients after RT.

In addition to the genomics information, contrast-enhanced computed tomography (CECT), magnetic resonance imaging (MRI), and dose files routinely acquire clinical data for NPC patients with RT plans [22,23]. Radiomics and dosiomics are two quantitative information-extraction methods to provide minable texture and dose-distribution information for clinical prognosis prediction. Traditional experiences have demonstrated that single or double sources of data had limited prediction power for acute AOM. Integration of complementary data from multiple types of datasets can lead to an intricate outcome than a simple summation of information [24]. Integration of multimodal data from multiple sources, for instance, clinical, radiomics, and dosiomics for NPC patients, has the potential to overcome the boundaries of conventional medical analysis [25,26,27,28,29,30,31]. Unfortunately, few studies have reported the possibilities of radiomics or dosiomics for AOM prediction [13,32,33]. To the best of our knowledge, there are no studies in the literature assessing whether the data fusion of multi-regions and multimodalities could enhance their capability of severe AOM prediction.

In this study, we aimed to analyze multimodal data, including clinical, radiomics, and dosiomics information, to predict the incidence of severe AOM in NPC patients following RT/CRT. The radiomics and dosiomics data were extracted from multi-regions related to RT treatment. These data were also extracted from multimodalities of images (CECT, contrast-enhanced T1-weighted, and T2-weighted MRI). In daily clinical practice, clinicians could benefit from adjusting treatment plans before RT for patients with a higher possibility of developing severe AOM to achieve personalized diagnosis and treatment.

## 2. Materials and Methods

**Patient data.** All patient data were retrospectively collected from NPC patients who underwent CRT or RT at Hong Kong Queen Elizabeth Hospital from 2012 to 2015. Informed consent of patients was waived due to the nature of the retrospective study. NPC patients were recruited based on the following inclusion and exclusion criteria. The inclusion criteria were: (1) NPC patients with pathological validation and absence of distant metastasis and co-existing tumors of other types at diagnosis, (2) patients treated with a total RT dose of 60–70 Gy, and (3) patients with a completed set of clinical, image, and radiation dosimetry data. The exclusion criteria were: (1) patients aged less than 18, (2) patients without original image or clinical data; and (3) patients for whom exact standard terminology criteria for adverse events (CTCAE) evaluation for AOM had not been recorded. Symptoms in grades 1 and 2 were defined as mild AOM, and grades 3 and 4 as severe AOM [34] All the patients were negative of oral mucositis according to the CTCAE grading system before receiving radiation therapy.

Clinical variables included (1) treatment information: TNM stage, treatment, past health condition, allergy history, vision condition, hearing condition, and CTCAE evaluation for AOM and (2) demographic data: age, gender, body weight, height, body mass index (BMI), and smoking and drinking habits. All clinical variables were acquired one week before RT commencement, except the CTCAE evaluation results, which were recorded 4–5 weeks after RT commencement. The equation for BMI is defined as follows:Body mass index (BMI) = weight/height^2^(1)

Patients were maintained in a supine position during the imaging examination. Details of imaging acquisition are summarized in Table A1 and Table A2.

**Image pre-processing.** In this study, the imaging pre-processing steps were based on our previous work [28] and are in accordance with the Image Biomarker Standardisation Initiative (IBSI) guidelines [35]. Specifically, (1) voxel size resampling: all images (CECT and MRI) were resampled to a voxel size of 1 × 1 × 1 mm^3^; (2) volume of interest (VOI) re-segmentation: CECT images were re-segmented to confine the Hounsfield unit (HU) to (–150,180) to eliminate the non-soft tissue in the VOI; (3) image filtering: a Laplacian of Gaussian (LoG) filter with three levels of Gaussian radius parameter was used under fine (1 mm), medium (3 mm), and coarse (6 mm) scales; (4) quantization of gray levels: gray- level intensities of the images were fixed to 50 bins; and (5) inhomogeneity correction of image pixel value: N4B bias correction in the “N4 Bias Field Correction Image Filter” in SimpleITK (v1.2.4) was implemented, in particular, to MRI images.

**Radiomics and dosiomics feature extraction.** Feature extraction was performed using our in-house platform based on publicly available SimpleITK (v1.2.4) and PyRadiomics (v2.2.0) [36,37]. All VOIs were delineated by an experienced senior clinician [38]. The gross tumor volumes (GTVs) were contoured based on CECT with the assistance of MRI images.

**Radiomics.** The gross tumor volume of the NPC primary tumor (GTVnp) and the gross tumor volume of nodal lesions (GTVn) were selected as the main VOIs for radiomics feature extraction. Features with or without LoG filters were both involved. All these features were extracted from CECT, contrast-enhanced T1 weighted (cT1WI), and T2 weighted (T2WI) images (for details, please refer to Figure 1). Meaning of each VOI for different image modalities were listed in Table 1.

Three categories of radiomics features were extracted: shape, first-order statistics, and texture features. The texture features can be further categorized into gray-level difference matrix (GLDM), gray-level cooccurrence matrix (GLCM), gray-level run-length matrix (GLRLM), gray-level size-zone matrix (GLSZM), and neighboring gray-tone difference matrix (NGTDM) classes.

**Dosiomics.** Except for GTVnp and GTVn, the region of the high-dose nodal planning target volume (PTVn_70Gy) and region of the low-dose nodal planning target volume (PTVn_60Gy) were also added to the dosiomics analysis (please refer to Figure 1, Figure A2 and Table 1 for more details).

Two-dimensional (2D) and three-dimensional (3D) dose–volume histograms (DVHs) of each studied VOI were extracted from dose files for dosiomics feature extraction. All dosiomics features were extracted based on Gabry et al.’s previous study [39]. Features that reflect dose distribution, for instance, mean dose, spatial dose gradient, and spatial dose spread were extracted accordingly. All the calculation algorithms have been listed in a previous publication by Buettner et al. [40].

**Model development and evaluation.** The statistical analysis, model training, and evaluation were conducted in Jupyter 6.4.12 and SPSS 25. The receiver operating characteristic (ROC) curve and area under the ROC curve (AUC) with 10-fold validation was performed to evaluate model performance. The CTCAE grade scale of patients in mucositis was dichotomized between severe AOM (grade ≥ 3) and mild AOM (grade ≤ 2) as the prediction endpoint. Patients were stratified based on CTCAE grade to training and testing groups at a 7:3 ratio (details in Figure 2).

**Single-modal models.** The data sources for single-modal models were restricted to single modality of data (radiomics, dosiomics, or clinics), single modality of imaging (CECT, cT1WI, T2WI, and DVH), and single region of patients (GTVnp, GTVn, PTVn_60Gy, and PTVn_70Gy). Each single data set had two steps in this stage: (1) feature selection and model training in the training group set and (2) AUC evaluation in validation groups.

For clinical data, chi-square and Mann–Whitney U tests were employed for binary and non-binary variables for univariate analysis. *p* values < 0.05 was considered to be statistically significant. All radiomics and dosiomics data were standardized with the MinMax scaler before selection. For radiomics and dosiomics data, we first identified significant features between severe and mild AOM patients in the training set with Mann–Whitney U tests. After that, random forest (RF) was used to rank the importance of the significant features considering both feature interactions and nonlinearities. The optimal feature number was set according to the best RF training model score. Three models, including logistic regression (LR), Gaussian Naïve Bayes (GNB), and extreme gradient boosting (XGBoost), were applied to evaluate the combined predictive value of these selected features in the independent validation set. All VOIs data were analyzed separately at the single model stage.

**Multimodal data integration**. Clinical data after multivariant analysis (LR) with *p* value < 0.05 were selected for data integration. Dosiomics and radiomics data from different VOIs and image modalities were integrated with two methods: (1) dosiomics and radiomics data were combined together before feature selection and (2) the features selected from the RF model were merged and directly combined without a further feature-selection step (please refer to Figure 2 for more details). All the data-integration methods are listed in Table 2. 

Shapley Additive Explanations (SHAP), an explainable artificial intelligence (AI)-based tool, was applied for further explanation of feature importance for the model with the best AUC result and specific features [41].

## 3. Results

### 3.1. Patient Characteristics

A total of 397 continuous patients were collected based on their final diagnosis with pathological validation. Of these patients, with a median age of 54 (range 26–86 years), 242 were enrolled for further analysis following the inclusion and exclusion criteria (details in Figure A1). All patients were negative for oral mucositis with CTCAE graded 0 before radiation therapy. Univariate analysis results of demographic and clinical characteristics for those patients are listed in Table 3.

### 3.2. Feature Extraction and Model Development

#### 3.2.1. Feature Extraction

In this study, a total of 1544 radiomics features, 386 features each for four modalities of imaging, were extracted from raw and LoG-filtered images. A total of 836 dosiomics features (210 for GTVn, 211 for GTVnp, 204 for PTVn_60Gy, and 211 for PTVn_70Gy) were extracted from dose images.

#### 3.2.2. Models

For the clinical data, four variables, including age, RT treatment alone, T stage, and smoking habits, were selected after univariate analysis. The logistic regression (LR) model was established with these variables. T stage and smoking habits had statistical significance in the LR model with a *p*-value < 0.05 (details in Table 4).

Radiomics and dosiomics features extracted from various VOIs were put into Mann–Whitney U tests and RF classifier step by step. RF selection results of the threshold and feature numbers are listed in Table A3.

Nine categories of single-modal models (C, PTVn_70Gy_D, PTVn_60Gy_D, GTVn_D, GTVnp_D, GTVn_R, GTVnp_R_T2, GTVnp_R_CECT, and GTVnp_R_cT1) were established with single modal, single modality, and single VOI data. The best validation AUC was at 0.75 ± 0.12 (training AUC = 0.73 ± 0.01) of a GNB model (GTVnp_R_cT1) with radiomics data from GTVnp of cT1WI. Seven groups of models with data integrated before feature selection (raw-data integration) were generated with the best AUC of a GNB model (GTVnp_RD) at 0.81 ± 0.01 (training AUC = 0.79 ± 0.01). This best-performing model was constructed with features selected from radiomics and dosiomics data in the region of GTVnp. In addition, six sets of combined data after feature selection were also used for modeling. A best LR model (C&R&D) with AUC at 0.79 ± 0.14 (training AUC = 0.81 ± 0.02) was set with the simply combined data of selected clinical, dosiomics, and radiomics features (details of mean 10-fold validation AUC results are listed in Figure 3).

The SHAP analysis showed the importance of the five features in the GTVnp_RD model for prediction of severe AOM. Four of the five features were derived from cT1WI. All five features are texture features. No dosiomics features were selected after the feature selections (details in Figure 4).

## 4. Discussion

In our study, we used simply combined and data-fusion methods to manage multimodalities of data (clinical, radiomics, and dosiomics), multimodalities of imaging (CECT, cT1WI, and T2WI), and multi-regional information (GTVn, GTVnp, and PTVn) to predict the incidence of severe AOM. Multiple models were established to evaluate and determine which method was effective for clinical decision-making. Comparison of the AUC between models showed that the simple combination of single-modal data of selected features had the most stable performance (C&R&D), with an average AUC of 0.77 ± 0.17. In addition, data-fusion methods, integrating radiomics and dosiomics data before selection procedures, resulted in the best-performing model (GTVnp_RD), with the best test AUC of 0.81 ± 0.01. This is also the best AUC among the existing AOM prediction models from previous studies.

The feature numbers in the C&R&D model and GTVnp RD were 29 and 5, respectively. Obviously, data fusion was more efficient for training a model with one-sixth the number of features to achieve stronger model predictability. To better explain the correlations of the selected features and severe AOM for NPC patients, a SHAP plot was applied for the GTVnp_RD XGBoost model. In this model, radiomics features extracted from GTVnp in cT1WI images yielded the highest and majority prediction value for severe AOM.

Poolakkad and his colleagues established a machine learning (ML) model of 253 H&N patients’ clinical data with the best AUC of 0.79 for AOM prediction [42]. Most clinical data selected in their study were late after the CRT scheme, for example, the anti-neoplastic chemotherapy-induced pancytopenia, co-morbidity score, and agranulocytosis. It is worth noting that the features and variables selected in our study were all from the data collected before implementation of the RT regimen. Clinicians could predict the severe AOM before the commencement of RT planning. Personalized treatment strategies adjustment could be achieved using the developed prediction model.

Strictly speaking, the concept of dosiomics is originated from radiomics. The data for dosiomics and radiomics are similar in terms of the feature calculation algorithm [43]. The clinical data are different from the “-omics” data in nature. Therefore, instead of integrating raw clinical data with other data, the combination of selected clinical data was only applied in this study. Compared with less increase of AUC for the combination of clinical data with integrated RD data, the clinical data could enhance the prediction capability of single-modal models. The single radiomics models (R) and single clinical models (C) have limited prediction performance with average AUC of three models (LR, GNB, and XGBoost) at 0.63 ± 0.06 and 0.63 ± 0.64, respectively. When combining the clinical data with the selected radiomics features, the model (C&R) outperformed both R and C models with average AUC at 0.74 ± 0.03.

The single dosiomics models yielded poor performance, with most AUCs under 0.7 in the validation data set. In a previous study, dose distribution correlated with the incidence and severity of AOM [44]. Dean et al. [45] developed an RF model with a testing AUC of 0.71 ± 0.09, using a dose–volume histogram, spatial dose metrics from the oral cavity, and clinical data. In the current study, the best dosiomics model had the mean testing AUC of 0.69 ± 0.14. Different tumor-related VOIs may present different prediction value for severe AOM. The difference in VOI selection between the two studies might shed some light on the discrepancy in the findings. The oral cavity directly represents the dose distribution in the oral mucosa, which might be more accurate than the GTVn, GTVnp, PTVn 60 Gy, or PTVn 70 Gy. The VOI of the oral cavity requires specific contouring. It is worth noting that contouring of the oral cavity is not a common practice in the participating hospital of this study. Extra contouring is labor-intensive work in daily clinical practice. Our study only selected the routine VOI broadly used for RT planning, which could support our model to be applied from bench to bedside for clinical decision-making. Besides, the DVH is prone to over-simplifying the dose distribution [46]. It is recommended to combine or integrate dosiomics data with other modalities of data. When incorporating dosiomics data with other data types, the best mean validation AUC could surge to 0.81 ± 0.01.

At present, there exists no effective preventive measures for the occurrence of severe AOM in NPC patients undergoing RT. Nevertheless, it is feasible to mitigate the severity of this affliction: (1) Use of alternative radiation techniques, such as proton therapy, may be considered to reduce the risk of oral mucositis while maintaining treatment efficacy [34,47]. (2) Shortening the duration of chemotherapy. For advanced NPC patients who need to accept both radiotherapy and chemotherapy, shortening the exposition time to chemotherapy agents has shown lower mucosal toxicity [48]. (3) Photobiomodulation is a supportive treatment for the protection of high-risk mucositis patients [49]. (4) Supportive care interventions: preemptive or proactive use of supportive care interventions, such as oral hygiene measures, pain management, or nutritional support, may be considered to prevent or reduce the severity of AOM [50].

The limitations of our study were: (1) The mucositis grade levels of our patients had an imbalanced distribution. This might have had a negative influence on the data analysis work. The imbalanced results were the nature of the clinical situation. Patients were stratified into the training and validation groups according to the severity of OM, which could offset the imbalance problem [18,19,34]. (2) Potential bias of smoking information: in our study, the number of smoking patients might be underestimated due to the nature of this patient-reported outcome. This data were reported by patients at the time of their hospital visit and recorded in the nursing consultation notes. (3) The severity of AOM was scaled with standard terminology criteria for adverse events (CTCAE) in v3 or v4.03, almost equivalent to mucositis. Various criteria are available for mucositis grading, such as the those of the Radiation Therapy Oncology Group (RTOG) and the World Health Organization (WHO). These scales have excellent concordance with bundled scores of 3 and 4 to describe severe AOM [51]. The CTCAE is easily conducted by clinicians and nurses and broadly applied in the hospital. (4) The correlations of contributors under AOM for NPC patients are complex. For clinical decision-making, genome information, other clinical information such as fermented-food consumption and EBV infection, and pathological image may also play critical roles. The limited data resources for multimodal data integration are common challenges in the data-mining field. The radiomics data in our study also provided relevant genomic information. Compared with gene test results, the CECT and MRI examination images collected in our research are clinical routines used by clinicians to set the RT plan for NPC patients. These noninvasive examinations could serve as high-throughput screening tools for further application of severe AOM prediction in the future. (5) Other selection of VOIs: for practical consideration, we have not added the VOIs of the oral cavity, tongue, pharyngeal muscles, etc., which may hold potential predictive value for AOM. Further investigation is recommended to incorporate this information to enhance the accuracy of the analysis.

## 5. Conclusions

AOM is a challenging and distressing complication in NPC patients following RT. Prediction of severe AOM is necessary for timely prevention and intervention, which would further improve the QoL and survival of patients. In this study, we adopted multimodal data (clinical, radiomics, and dosiomics), multimodality of imaging (CECT, cT1WI, and T2WI), and multi-regional information (GTVn, GTVnp, and PTVn) to develop a best-performance model for severe AOM prediction. The simple combination of selected information and data fusion were applied in our work. The results demonstrated that the fusion of radiomics and dosiomics data from the primary tumor could generate the most effective and best-performing model (mean AUC = 0.81 ± 0.01). The data resources and VOIs selected in this study are routinely used in clinical practice, which has excellent potential for further clinical support. Further validation work on a large cohort is warranted to validate model generalizability.

## Figures and Tables

**Figure 1 cancers-15-02032-f001:**
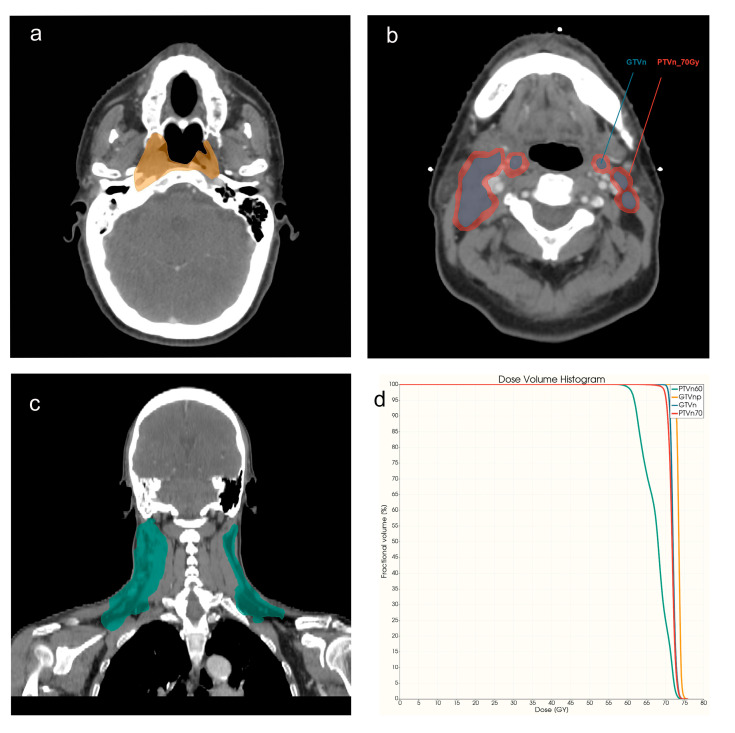
VOI examples for NPC patients with CECT examination. (**a**) Region of GTVnp (orange), axial view. (**b**) Region of GTVn (blue) and PTVn_70 Gy (red), axial view. (**c**) Region of PTVn_60 Gy (green), coronal view. (**d**) DVH curve of four VOIs.

**Figure 2 cancers-15-02032-f002:**
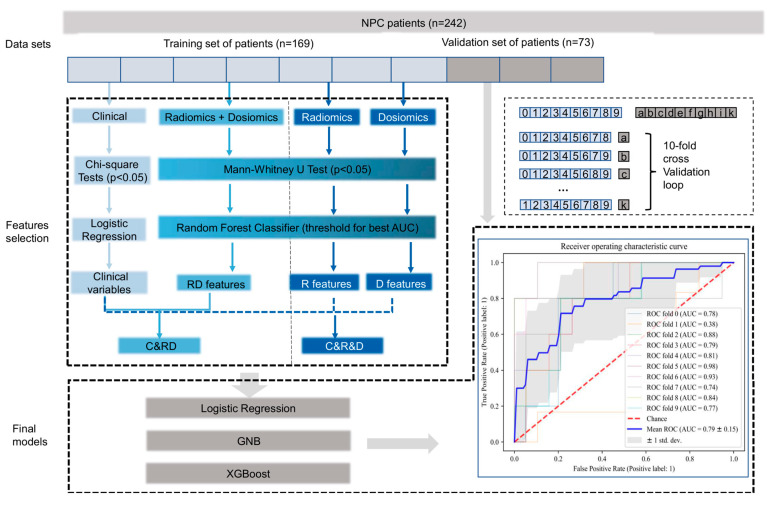
Scheme of feature selection and modeling. Training and validation sets were separated before data analysis. The training set of data was used for feature selection. The validation set of data was used for model evaluation. To further manipulate the numerical and categorical data, reduce the interactions, and solve the collinearity problems, random forest (RF) selection was applicated for radiomics, dosiomics, and integrated data. Three linear or non-linear models were developed with independent validation data sets with selected features. The area under the curve (AUC) was set as the main evaluation method for the model performance.

**Figure 3 cancers-15-02032-f003:**
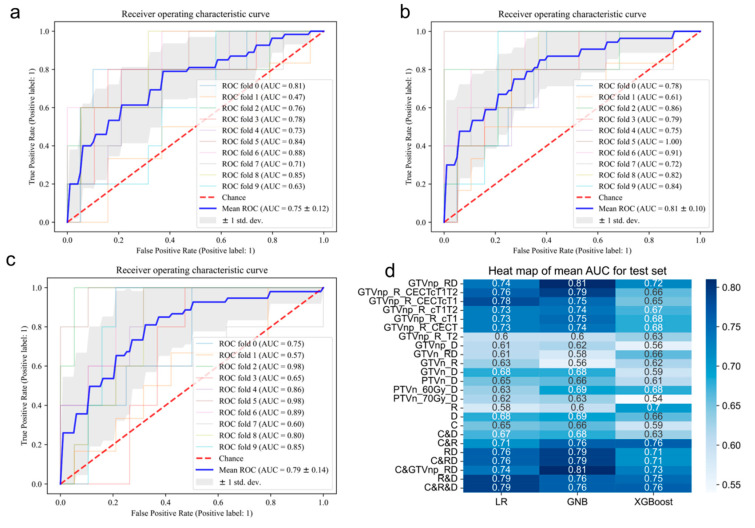
10-fold validation AUC results for the test set. (**a**) The AUC plot of GNB model for the GTVnp_R_cT1 data set. (**b**) The AUC plot of GNB model for the GTVnp_RD data set. (**c**) The AUC plot of LR model for the C&R&D data set. (**d**) The heatmap of mean AUC results for all models.

**Figure 4 cancers-15-02032-f004:**
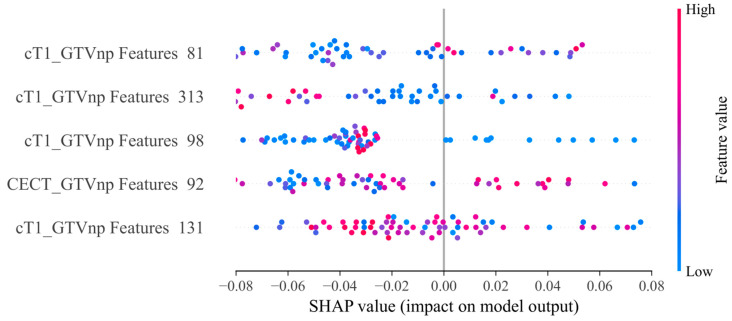
Feature importance of SHAP for XGBoost model of GTVnp_RD. From the highest to the lowest level, the features are categorized in GLSZM, log sigma 60 mm 3D GLCM, original GLDM, GLDM, and log sigma 10 mm 3D GLCM.

**Table 1 cancers-15-02032-t001:** VOIs and image modalities.

VOIs	Descriptions of VOI	Imaging Modalities/Images
GTVnp	Gross tumor volume of primary NPC tumor	CECT, cT1WI, T2WI, DVH
GTVn	Metastatic lymph nodes area	CECT, DVH
PTVn_70Gy	Regions of nodal planning target volume with	DVH
	the prescribed dose level of 70Gy	
PTVn_60Gy	PTVn with the prescribed dose level of 60Gy	DVH

**Table 2 cancers-15-02032-t002:** Data resources and integration/combination methods.

Name of Model	Methods
GTVnp_RD	Integration of radiomics and dosiomics GTVnp data before feature selection

GTVnp_R_CECTcT1T2	Integration of radiomics GTVnp data from CECT, cT1WI, T2WI before feature selection

GTVnp_R_CECTcT1	Integration of radiomics GTVnp data from CECT and cT1WI before feature selection

GTVnp_R_cT1T2	Integration of radiomics GTVnp data from cT1WI and T2WI before feature selection

GTVnp_R_cT1	Single radiomics data from CT1WI
GTVnp_R_CECT	Single radiomics data from CECT
GTVnp_R_T2	Single radiomics data from T2WI
GTVnp_D	Single dosiomics data from GTVnp
GTVn_RD	Integration of radiomics and dosiomics data from GTVn before feature selection

GTVn_R	Single radiomics data from GTVn
GTVn_D	Single dosiomics data from GTVn
PTVn_D	Integration of 60 and 70 Gy dosiomics data before feature selection
PTVn_60Gy_D	Single dosiomics data from PTVn_60Gy
PTVn_70Gy_D	Single dosiomics data from PTVn_70Gy
R	Integration of all radiomics data before feature selection
D	Integration of all dosiomics data before feature selection
C	Single clinical data
C&D	Combine selected clinical and dosiomics data for modeling
C&R	Combine selected clinical and radiomics data for modeling
RD	Integration of radiomics and dosiomics data before feature selection
C&RD	Combine selected clinical and RD data for modeling
C&GTVnp RD	Combine selected clinical and GTVnp RD data for modeling
R&D	Combine selected radiomics and dosiomics data for modeling
C&R&D	Combine selected clinical, radiomics and dosiomics data for modeling

**Table 3 cancers-15-02032-t003:** Demographic and clinical characteristics for all patients.

Characteristics	AOM < Grade 3 (Mild AOM)	AOM ≥ Grade 3 (Severe AOM)	*p* Value

Total Number	191 (78.9%)	51 (21.1%)	
Age, mean ± SD, years	54.89 ± 12.25	50.9 ± 10.60	0.036 *
18–65	149 (61.6%)	44 (18.1%)	
≥65	42 (17.4%)	7 (2.9%)	0.192
Gender			
Male	135 (55.8%)	41 (16.9%)	
Female	56 (23.1%)	10 (4.1%)	0.167
Treatment			0.004 *
RT alone	27 (11.2%)	0	
CRT	164 (67.8%)	51 (21.1%)	0.031 *
T stage			
T1	15 (6.2%)	3 (0.1%)	
T2	8 (3.3%)	5 (2.1%)	
T3	137 (56.6%)	28 (11.6%)	
T4	31 (12.8%)	15 (6.2%)	
N stage			0.091
N1	28 (11.2%)	1 (0.4%)	
N2	142 (58.7%)	45 (18.6%)	
N3	20 (8.2%)	5 (2.1%)	
Pathology			
Non-keratinizing squamous cell	175 (72.3%)	48 (19.8%)	0.556
Keratinizing squamous-cell carcinoma	16 (6.6%)	3 (1.3%)	0.487
Past health condition			
Past health good	92 (38.0%)	27 (11.2%)	
Basic diseases/cancer	99 (40.9%)	24 (9.9%)	0.545
Allegory of History			
No known drug allergies	176 (72.7%)	46 (19.0%)	
Allergy history	15 (6.2%)	5 (2.1%)	0.653
Vision			
Normal	189 (78.1%)	51 (21.1%)	
With eye impairment	2 (0.8%)	0	0.463
Hearing			
Normal	186 (76.9%)	48 (19.8%)	
With hearing impairment	5 (2.1%)	3 (1.2%)	0.247
Habits			
Smoking	9 (3.7%)	6 (2.5%)	0.044 *
Non-smoker	182 (75.2%)	45 (18.6%)	
Drinking	4 (1.7%)	1 (0.4%)	
No alcohol consumption	187 (77.3%)	50 (20.7%)	0.953
Height, mean ± SD, cm	163.4 ± 8.5	165.0 ± 8.0	0.561
Body weight, mean ± SD, kg	63.1 ± 11.9	66.2 ± 14.6	
1st week of RT			1.599
2nd week of RT	62.0 ± 11.8	64.9 ± 14.5	1.5
3rd week of RT	61.2 ± 11.4	63.9 ± 14.1	0.116
4th week of RT	60.2 ± 11.3	62.8 ± 14.0	1.418
BMI			
1st week of RT			
<25	131 (54.1%)	32 (13.2%)	
≥25	60 (24.8%)	19 (7.9%)	0.429
2nd week of RT			
<25	131 (54.1%)	51 (21.1%)	
≥25	60 (24.8%)	22 (9.1%)	0.116
3rd week of RT			
<25	131 (54.1%)	31 (12.8%)	
≥25	55 (22.7%)	20 (8.3%)	0.153
4th week of RT			
<25	142 (58.7%)	34 (14.0%)	
≥25	49 (20.2%)	17 (7.0%)	0.274

* *p* < 0.05. All the above data are derived from biopsy-proven primary NPC patients without the existence of distant metastasis or co-existing tumors of other type at diagnosis.

**Table 4 cancers-15-02032-t004:** Logistic regression results for single clinical data model.

Variables	*p*-Value	95% Confidence Interval	
		Lower 95% Bound	Upper 95% Bound
Age (18, 65)	0.802	0.345	2.274
T	0.007 *		
T 1	0.591	0.149	2.96
T 2	0.069	0.881	29.854
T 3	0.024 *	0.195	0.891
RT alone	0.998	0	.
Smoker	0.043 *	1.037	10.683

* *p* < 0.05.

## Data Availability

For patients’ privacy protection, the data availability was not applicable.

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
