# Peer review of "Multimodal Data Integration to Predict Severe Acute Oral Mucositis of Nasopharyngeal Carcinoma Patients Following Radiation Therapy"

_cancers, 2023, doi:10.3390/cancers15072032_

Round 1

Reviewer 1 Report

Brief summary

The paper focuses the attention on an impairing complication that can follow radiation therapy used in treatment of nasopharingeal carcinoma (NPC), which is acute oral mucositis (AOM).

This is a clear paper, with structured and solid analysis of different radiomics and dosiometric factors that can be used to build a predictive model to select AOM susceptible patients and to help clinicians in therapeutical choices.

The results are reproducible, and the conclusions are consistent with the thesis and argument presented.

The conclusions are interesting and add advances in the current knowledge of this radiotherapy side effect.

No ethical problems are found in this study

However, some criticisms are present in the paper.

General concept comment

The topic is interesting and original, as only few other complications of radiotherapy used for NPC are analysed in literature, as temporal lobe injury, salivary amount reduction and cervical spine osteoradionecrosis. The aim of the majority of these studies is similar to the purpose of this paper, which is to create models to predict the occurrence and severity of these conditions, in order to find susceptible patients and prevent radiotherapy side effects.

Oral mucositis occurring in the setting of radiotherapy for NPC is an impairing condition, and only few predictive models are present in literature. The main factors studied are clinical parameters and dosiometric parameters, while the combination of radiomics and dosiometric features is original and appreciable (Li PJ, Li KX, Jin T, Lin HM, Fang JB, Yang SY, Shen W, Chen J, Zhang J, Chen XZ, Chen M, Chen YY. Predictive Model and Precaution for Oral Mucositis During Chemo-Radiotherapy in Nasopharyngeal Carcinoma Patients. Front Oncol. 2020 Nov 5;10:596822. doi: 10.3389/fonc.2020.596822.)  The combination of dosiomics and radiomic features in this paper  is

However, some points have to be clarified

-        English language should be revised, as some sentences are too complex, long or articulate.

-        Materials and methods section

o   Please specify that the patients included in the study have no distant metastatic disease, as in the paper you also described dosiometric and radiomic features in nodal disease but in line 83 one of the inclusion criteria in “NPC with pathological validation and without metastasis”, while nodal localisation of disease is actually a metastasis

o   In clinical variables no information is given on histological subtypes of NPC. It should be specified if all cases are non-keratinising squamous cell carcinoma, otherwise you should classify NPC following the 4th WHO Classification of Head and Neck tumor (2017) - Nonkeratinizing squamous cell carcinoma (differentiated and undifferentiated subtype; Keratinizing squamous cell carcinoma; Basaloid squamous cell carcinoma -

o   In the present study, AOM severity was scored using CTCAE method, which is also cited as one of the limitation factors of the paper. In literature there are different methods for the evaluation of severity of AOM, as Toxicity criteria of the Radiation Therapy Oncology Group (RTOG), the European Organization for Research and Treatment of Cancer (EORTC), and the criteria set out by the World Health Organization (WHO) and the OM Assessment Scale. Please explain the motivations of your choice. Moreover, I would recommend trying to re-assess the results using different scoring systems, even by comparing them to CTCAE (see the methods above)

o   No data are available regarding CTCAE at the time of NPC diagnosis, as you specify in this part of the paper. It should be clarified if all the patients were negative for oral mucositis before radiation therapy.

o   Non keratinising squamous cell carcinoma of nasopharynx is more commonly relate to fermented food and/or salted fish consumption (high notrosamine or nitrosamine precursor content) and EBV infection especially in endemic region, while HPV present a risk factor especially in non endemic regions. It would be appreciable to add at least EBV infection, if you have no information of personal food habits, as a risk factor in method section and table 3

Specific comments:

- In the title, “acute mucositis” should be modified in “severe acute oral mucositis” or “acute oral mucositis” as the localisation of the acute inflammation herein studied regards oral cavity in the paper

- Through the different section of the paper, OM (oral mucositis) should be changed in acute oral mucositis “AOM” 

- In the different parts of the paper, is not clear if you analysed only severe OAM or even the other grade of the disease, while it is well described in Table 3. Please add more information about this topic in materials and method, results and discussion sections 

- Line 21: “over 95% of NPC are non-keratinising", please specify non keratinising squamous cell carcinoma, and change subtypes into subtype. It would be appreciable to add a reference for the epidemiologic data you write here (Es: - Chang ET, Ye W, Zeng YX, Adami HO. The Evolving Epidemiology of Nasopharyngeal Carcinoma. Cancer Epidemiol Biomarkers Prev. 2021 Jun;30(6):1035-1047. doi: 10.1158/1055-9965.EPI-20-1702. Epub 2021 Apr 13)

- If you have it, you can add clinical pictures of acute oral mucositis

- If you have it, you can add histological pictures of NPC, with the different subtypes based on the presence of them in the cohort herein studied

-line 52: SNP is cited for the first time, please add the complete meaning (Single nucleotide polymorphism)

- table3:

o   Please check that percentages are correct, e.g.: in Gender and Treatment the numbers do not sum up to 100%

o    Please check significant figures through this table and the paper, e.g., in Height and Weight there is no need for 2 figures after the comma, since it would exceed the precision of the instrument used to measure those values. Example: 63.06±11.86 would become 63.1 ± 11.9kg, or even 63 ± 12kg, depending on the precision of the scale.

o   please specify the total number of patients with OM grade < 3 (191) and OM grade > 3 (51) in the first line of the table

o   Please specify the number of non keratinising, keratinising and basaloid NPC

o   Please describe the types of basic disease/cancer in a note at the end of the table

o   I would change “none smoking” and “none drinking” with “no smoke” and “no alcohol consumption”; moreover fermented food and/or salted fish consumption and EBV infection as a risk factors for non kearinizing NPC

Villa A, Vollemans M, De Moraes A, Sonis S. Concordance of the WHO, RTOG, and CTCAE v4.0 grading scales for the evaluation of oral mucositis associated with chemoradiation therapy for the treatment of oral and oropharyngeal cancers. Support Care Cancer. 2021 Oct;29(10):6061-6068. doi: 10.1007/s00520-021-06177-x.

In conclusion I suggest that the paper should be reconsidered after major revisions

Author Response

The topic is interesting and original, as only few other complications of radiotherapy used for NPC are analysed in literature, as temporal lobe injury, salivary amount reduction and cervical spine osteoradionecrosis. The aim of the majority of these studies is similar to the purpose of this paper, which is to create models to predict the occurrence and severity of these conditions, in order to find susceptible patients and prevent radiotherapy side effects.

Oral mucositis occurring in the setting of radiotherapy for NPC is an impairing condition, and only few predictive models are present in literature. The main factors studied are clinical parameters and dosiometric parameters, while the combination of radiomics and dosiometric features is original and appreciable (Li PJ, Li KX, Jin T, Lin HM, Fang JB, Yang SY, Shen W, Chen J, Zhang J, Chen XZ, Chen M, Chen YY. Predictive Model and Precaution for Oral Mucositis During Chemo-Radiotherapy in Nasopharyngeal Carcinoma Patients. Front Oncol. 2020 Nov 5;10:596822. doi: 10.3389/fonc.2020.596822.)  The combination of dosiomics and radiomic features in this paper is However, some points have to be clarified

-        English language should be revised, as some sentences are too complex, long or articulate.

Response: Thank you very much for taking time to provide us with valuable feedbacks. We have thoroughly revised the language, simplified some long and complex sentences in the revised manuscript. As it might be challenging to copy all the revisions in this response letter, we would like to kindly refer you to the revised manuscript, changes are made with track changes.

-        Materials and methods section

o     Please specify that the patients included in the study have no distant metastatic disease, as in the paper you also described dosiometric and radiomic features in nodal disease but in line 83 one of the inclusion criteria in “NPC with pathological validation and without metastasis”, while nodal localisation of disease is actually a metastasis

Response: We thank the reviewer for this careful comment, and apologize for causing confusion. We would like to clarify that the patients included in this study presented absence of distant metastasis and co-existing tumors of other types at diagnosis, while patients may present nodal metastatic diseases (referred to as N1, N2, or N3 stage). We have now revised the relevant information in the revised manuscript. Particularly, the concerned sentence has now been changed to “NPC with pathological validation and absence of distant metastasis and co-existing tumors of other types at diagnosis”. (Please refer to Line 83 and Table 1 of the revised manuscript for more details)

 o  In clinical variables no information is given on histological subtypes of NPC. It should be specified if all cases are non-keratinising squamous cell carcinoma, otherwise you should classify NPC following the 4th WHO Classification of Head and Neck tumor (2017) - Nonkeratinizing squamous cell carcinoma (differentiated and undifferentiated subtype; Keratinizing squamous cell carcinoma; Basaloid squamous cell carcinoma –

Response: Thank you for your thoughtful comments. We have now specified the histological subtypes of NPC in the included patients, according to the WHO classification. Please refer to Table 3 of the revised manuscript and the response to the second specific comments in this letter for more details.

o      In the present study, AOM severity was scored using CTCAE method, which is also cited as one of the limitation factors of the paper. In literature there are different methods for the evaluation of severity of AOM, as Toxicity criteria of the Radiation Therapy Oncology Group (RTOG), the European Organization for Research and Treatment of Cancer (EORTC), and the criteria set out by the World Health Organization (WHO) and the OM Assessment Scale. Please explain the motivations of your choice. Moreover, I would recommend trying to re-assess the results using different scoring systems, even by comparing them to CTCAE (see the methods above)

Response: We thank the reviewer for raising another thoughtful comment. We acknowledge that various criteria are available for mucositis grading. There are several compelling reasons that we employed CTCAE scoring system in this study, as described below:

  1. First, the CTCAE method was adopted in the participating hospital to evaluate the grading of oral mucositis of the NPC patients who were admitted to radiation therapy at their center. Due to the retrospective nature of this study, this is the eligible data available to us at the time we prepared this manuscript, and we were unfortunately unable to request the physicians to re-assess every histologic patient, who has already gone through the treatment, using other grading systems, especially given their existing heavy clinical workload.
  2. Second, the CTCAE method is one of the most widely adopted scoring system for mucositis within the community. We believe that the adoption of the CTCAE method in this study would favor comparisons between results of this study and those (to be) reported in the literature.

In addition, the RTOG, WHO and CTCAE grading systems may lack of concordance for grading mild to moderate AOM (grade 1 to 2), while it shows a high degree of concordance of results when using WHO grade 3 or 4, RTOG grade 3 or 4, and CTCAE grade 3 or 4 to describe the severe mucositis grade.(  In our study, we bundled grade  3 and grade 4 to represent severe AOM, which have been indicated to have excellent concordance among CTCAE, RTOG and WHO.[1] Therefore, we hope that the reviewer could show his/her generous understanding on the underlying rationales of using the CTCAE system alone in the present work. We have added relevant explanations in the Discussion Section of the revised manuscript. (Line 271)

o   No data are available regarding CTCAE at the time of NPC diagnosis, as you specify in this part of the paper. It should be clarified if all the patients were negative for oral mucositis before radiation therapy.

Response: Thank you for the valuable suggestions. We have now clarified in the Methods Section that all the patients were negative of oral mucositis according to the CTCAE grading system before receiving radiation therapy. Please refer to Line 89 and 164 of the revised manuscript for more details.

o   Non keratinising squamous cell carcinoma of nasopharynx is more commonly relate to fermented food and/or salted fish consumption (high notrosamine or nitrosamine precursor content) and EBV infection especially in endemic region, while HPV present a risk factor especially in non endemic regions. It would be appreciable to add at least EBV infection, if you have no information of personal food habits, as a risk factor in method section and table 3.

Response: Thank you for your thoughtful comments. We would like to refer the reviewer to our detailed response in the last part of the 4th specific comments in this letter.

Specific comments:

Point 1: In the title, “acute mucositis” should be modified in “severe acute oral mucositis” or “acute oral mucositis” as the localisation of the acute inflammation herein studied regards oral cavity in the paper

-Through the different section of the paper, OM (oral mucositis) should be changed in acute oral mucositis “AOM”

-In the different parts of the paper, is not clear if you analysed only severe OAM or even the other grade of the disease, while it is well described in Table 3. Please add more information about this topic in materials and method, results and discussion sections

Response 1: Thank you for your valuable feedback.

-The title has been changed from “acute mucositis” to “severe acute oral mucositis”.

-The expression of OM in the article have been modified to “AOM”.

- We understand that it was not clear in certain parts of the paper whether we analyzed only severe AOM or all grades of the disease. We would like to clarify that we developed a machine learning model to distinguish severe AOM patients from mild AOM patients. For this purpose, we divided the patients into two groups: grade 1-2 (representing mild AOM patients) and grade 3-4 (representing severe AOM patients). We have added more information regarding this topic in the materials and methods, results, and discussion sections, as suggested. (Please refer to line 88, line 272, and the title line of Table 3 for the relevant descriptions)

Point 2: Line 21: “over 95% of NPC are non-keratinising", please specify non keratinising squamous cell carcinoma, and change subtypes into subtype. It would be appreciable to add a reference for the epidemiologic data you write here (Es: - Chang ET, Ye W, Zeng YX, Adami HO. The Evolving Epidemiology of Nasopharyngeal Carcinoma. Cancer Epidemiol Biomarkers Prev. 2021 Jun;30(6):1035-1047. doi: 10.1158/1055-9965.EPI-20-1702. Epub 2021 Apr 13)

- If you have it, you can add clinical pictures of acute oral mucositis

- If you have it, you can add histological pictures of NPC, with the different subtypes based on the presence of them in the cohort herein studied

Response 2: Thank you for your detailed suggestions. We have made the necessary changes in the revised manuscript, as indicated as follows:

-The “non-keratinising” has been changed to “non-keratinising squamous cell carcinoma” and “subtypes” to “subtype”. The suggested reference has been added. (Line 21)

- We agree that incorporating clinical pictures of AOM and histological pictures of NPC into the manuscript would enhance readers' understanding of AOM. However, due to ethical considerations, we have not obtained consents from patients to publish their personal pictures in the article. In the present work, we focused on analyzing features extracted from radiographic images (CT and MRI), radiation dose map, and text-based clinical attributes obtained from patient folders, and hope that the reviewer could understand our difficulty in obtaining the clinical or histological pictures for this publication purpose.

Point 3: line 52: SNP is cited for the first time, please add the complete meaning (Single nucleotide polymorphism)

Response 3: Thank you for your suggestion.

We have now added the complete meaning of single nucleotide polymorphism before using the term of SNP. (Line 52)

Point 4: table3:

o   Please check that percentages are correct, e.g.: in Gender and Treatment the numbers do not sum up to 100%

o    Please check significant figures through this table and the paper, e.g., in Height and Weight there is no need for 2 figures after the comma, since it would exceed the precision of the instrument used to measure those values. Example: 63.06±11.86 would become 63.1 ± 11.9kg, or even 63 ± 12kg, depending on the precision of the scale.

o   please specify the total number of patients with OM grade < 3 (191) and OM grade > 3 (51) in the first line of the table

o   Please specify the number of non keratinising, keratinising and basaloid NPC

o   Please describe the types of basic disease/cancer in a note at the end of the table

o   I would change “none smoking” and “none drinking” with “no smoke” and “no alcohol consumption”; moreover fermented food and/or salted fish consumption and EBV infection as a risk factors for non kearinizing NPC

Response 4:

Thank you for your thoughtful suggestions. We appreciate your attention to details and have made the necessary corrections to table 3:

-We have checked and revised the percentages in table 3;

-For the weight and height, the significant figures have been exacted to 1 figure;

-Total number of patients have been added in the first line of the table;

-We have now added the pathologic results in the table. (No patients with basaloid NPC were included in this study)

- We have now added that all the above data are derived from biopsy-proven NPC patients included in this study, at the end of the table;

- We have now changed the “none smoking” to “no smoke”, and “none drinking” to “no alcohol consumption” accordingly.

-We acknowledged that the fermented food consumption and EBV infection are risk factors for the incidence of non-keratinizing patients. There is a practical consideration that these data were not included in our analysis. As the information of consumption on fermented food and EBV infection were not described in detail or available for every NPC patient in the participating hospital, which has posed a challenge for us to collect such data and make meaningful comparisons between mild AOM and severe AOM patients while maintaining appropriate sample size for model development. Therefore, these data were not included in this study. Besides, the scope of this work focused on applying different data integration methods to multiple types of features from various tissues for predicting the incidence of severe AOM event (instead of the incidence of non-keratinizing patients). In this study, we have made our most efforts to include as much relevant clinical parameters as possible (as indicated in Table 3) and stipulated that omissions of these data (fermented food consumption and EBV infection) might have little impact on the conclusion of our findings. However, we are aware that the potential role of these data and have now put it as one of the limitations in the Discussion Section in this study. (Line 275)

Reference

  1. Villa, A.; Vollemans, M.; De Moraes, A.; Sonis, S. Concordance of the WHO, RTOG, and CTCAE v4.0 grading scales for the evaluation of oral mucositis associated with chemoradiation therapy for the treatment of oral and oropharyngeal cancers. Support Care Cancer 2021, 29, 6061-6068, doi:10.1007/s00520-021-06177-x.

Reviewer 2 Report

This article is a previous study of the integration of Ragiomics, Dosiomics, and clinical data as a tool to predict radiation mucositis in nasopharyngeal cancer patients in advance.

The method of data utilization is based on previous studies, and the methods of analysis are well supported and reliable. It is a new approach and I am impressed.

However, I have some serious questions.

1,

First, as a major revisit, the issue of smoking rates. The smoking rate in Hong Kong as of 2012-2015 was reported by a Hong Kong public agency to be about 20% for men and a few percent for women. The Centre for Health Protection Department of Health for disease prevention and control. In this paper, the smoking rate is clearly too low. There is a strong concern that the unclear‐smokers are assigned to nonsmokers. This is even stranger because nasopharyngeal cancer patients tend to have slightly higher smoking rates than healthy people. Since you have identified smoking as one important factor, it needs to be revisited and reanalyzed if necessary.

2

Second, the introductory section states that adjusting the treatment plan prior to RT will help achieve personalized diagnosis and treatment, but the discussion or summary does not provide specific suggestions on how to make adjustments. Suggestions for how this should be done should be included in the discussion or summary. This is a minor revision.

3

The dose to the oral cavity, including the tongue, is also important in radiation mucositis that makes treatment difficult, such as dysphagia, etc. We believe that you are treating patients with IMRT, but we think you should show some more dose maps for the usual NPC treatment at your hospital, including the oral cavity.

Author Response

This article is a previous study of the integration of Ragiomics, Dosiomics, and clinical data as a tool to predict radiation mucositis in nasopharyngeal cancer patients in advance.

The method of data utilization is based on previous studies, and the methods of analysis are well supported and reliable. It is a new approach and I am impressed.

However, I have some serious questions.

Point 1: First, as a major revisit, the issue of smoking rates. The smoking rate in Hong Kong as of 2012-2015 was reported by a Hong Kong public agency to be about 20% for men and a few percent for women. (The Centre for Health Protection ;Department of Health for disease prevention and control). In this paper, the smoking rate is clearly too low. There is a strong concern that the unclear‐smokers are assigned to nonsmokers. This is even stranger because nasopharyngeal cancer patients tend to have slightly higher smoking rates than healthy people. Since you have identified smoking as one important factor, it needs to be revisited and reanalyzed if necessary.

Response 1: Thank you for your valuable feedback. We appreciate your concern regarding the smoking rates in Hong Kong and their potential impact on the study's findings. We understand that the smoking rate reported by the Hong Kong public agency differs from the smoking rate reflected in our study. We acknowledge the presence of potential bias from patient-reported clinical data. This data was reported by patients at the time of their hospital visit and recorded in the nursing consultation notes where we retrieved the data. Due to retrospective design of this study, we may not be able to re-confirm this information with patients. We have been added a section in discussion section in this regard and reminded the readers to interpret this data with cautions. (Line 264)

Point 2: Second, the introductory section states that adjusting the treatment plan prior to RT will help achieve personalized diagnosis and treatment, but the discussion or summary does not provide specific suggestions on how to make adjustments. Suggestions for how this should be done should be included in the discussion or summary. This is a minor revision.

Response 2: Thanks you for your thoughtful comments. We have added a paragraph (Line 249) regarding possible actions which could be helpful for high-risk severe AOM patients. These actions include alternative radiation techniques, such as proton therapy; shortening the duration of chemotherapy; photobiomodulation; and supportive care interventions. We hope that these measures may be of assistance to at-risk NPC patients of severe AOM.

Point 3: The dose to the oral cavity, including the tongue, is also important in radiation mucositis that makes treatment difficult, such as dysphagia, etc. We believe that you are treating patients with IMRT, but we think you should show some more dose maps for the usual NPC treatment at your hospital, including the oral cavity.

Response 3: Thank you for taking the time to review our manuscript and for your valuable comments. We appreciate your insight and feedback on our study.

Regarding the dose to the oral cavity, we agree that it is one of the contributing factors in radiation mucositis. There are several reasons that hinder us to include dose of oral cavity in the present work:

  1. From practice considerations, the segmentation of oral cavity is not a routine clinical practice in RT planning in the participating hospital. Therefore, analyzing the impact of dose to oral cavity on the incidence of severe AOM in the present work would require additional segmentations from clinical oncologists, which may be practically challenging given their existing heavy clinical workload. From the application point of view, in the long run, we stipulated that models developed without inclusion of oral cavity features might be more applicable to other oncology centers where segmentation of oral cavity is not part of their routine clinical practice, therefore no additional segmentation works need to be conducted in order to apply the models in clinic.
  2. From the perspective of scope of this work, we have made an effort to include multiple types of imaging features (CECT, cT1WI MRI, and T2WI MRI) and dose-related features from primary NPC tumor and metastatic neck lymph nodes at different levels of prescription dose. We believe that these included datasets would allow us to develop valuable prediction models under the analyses of different data integration methods.

However, we agree that dose of oral cavity may also play a role in radiation mucositis, we have now put this as one of the limitations in this work and recommend researchers to include this information for future study. (Line 283) In addition, we have followed the suggestion of the reviewer to add a dose map of a representative patient in APPENDIX D.

Round 2

Reviewer 1 Report

Dear authors

the major revisions that I suggested have been taken into account and satisfied. 

Just few minor revisions:

- line 130:  "severe AOM (grade ≥ 3) and mild AOM (grade ≤ 2)" mild AOM should be ≤ 3, not 2 according to the table 3 

- in table 3 I would change "pathology" in "carcinoma" so that you can describe "keratinising SCC" and "non Keratinising SCC" (SCC squamous cell carcinoma have to be specify in the legend at the end of the table)

Reviewer 2 Report

The author's correction noted that this paper is a retrospective study and has limitations. Also, the correction I requested was executed.